# Evidence-Supported Automated Impressions for Alzheimer's Disease Detection from Brain MRI: A Feasibility Study

**Devesh Singh**[1] [iD]                        DEVESH.SINGH@MED.UNI-ROSTOCK.DE
[1] *Institute and Policlinic of Radiology, Pediatric Radiology and Neuroradiology, University Medical Center Rostock, Rostock, Germany*

**Dhanush H. Babu**[2]                        DHANUSH.BABU.HAREESH@FAU.DE
[2] *Friedrich-Alexander-Universität Erlangen-Nürnberg, Erlangen, Germany*

**Mathias Manzke**[1] [iD]                    MATHIAS.MANZKE@MED.UNI-ROSTOCK.DE
**Marc-André Weber**[1] [iD]            MARC-ANDRE.WEBER@MED.UNI-ROSTOCK.DE
**Majid Ramedani**[3] [iD]                          MAJID.RAMEDANI@DZNE.DE
[3] *German Center for Neurodegenerative Diseases (DZNE), Rostock, Germany*

**Arijana Bohr**[2] [iD]                            ARIJANA.BOHR@FAU.DE
**Emmanuelle Salin**[2] [iD]                      EMMANUELLE.SALIN@FAU.DE
**Björn Eskofier**[2] [iD]                        BJOERN.ESKOFIER@FAU.DE
**Martin Dyrba**[3] [iD]                            MARTIN.DYRBA@DZNE.DE

## Abstract

Convolutional Neural Networks (CNNs) are the standard models for neuroimaging analysis, but their opacity hinders clinical adoption. While Large Language Models (LLMs) offer a potential solution to translate CNN outputs as human-readable impressions, their reliability remains questionable. In the context of Alzheimer's Disease (AD) detection, we introduce a framework that computes CNN-based brain morphology scores and leverages a rule module for summarization. Using an LLM, conditioned on diagnostic guidelines via Retrieval-Augmented Generation (RAG), we generate explanatory justifications of pathology detected. To assess the impact of hallucinations, we propose a taxonomy that considers generated justifications as falsifiable claims. Manual evaluation on 30 reports shows that hallucinations remain substantial. In the pathological cases, from 34–50% of claims were incorrect. Our feasibility study shows that integrating rule-based guardrails with RAG improves auditability but fails to sufficiently mitigate hallucinations.

**Keywords:** LLM, Radiological Impressions, Dementia, MRI, RAG, Hallucination

## 1. Introduction

CNNs are widely used in medical imaging for diagnosing neurodegenerative diseases, and have shown strong performance in identifying pathological MRI patterns (Alsubaie et al., 2024). Correspondingly, LLMs offer promise in generating human-readable explanations (Zhou et al., 2025; Tian et al., 2023), but as they often hallucinate, their usability in clinical practice remains uncertain (Hager et al., 2024). In our feasibility study, we present a hybrid framework that integrates clinical domain knowledge with LLMs to generate radiological impressions of disease detection based on structural MRIs (see Figure 1).

Since no public dataset jointly contains MRI scans and corresponding radiologist reports, we first generated rule-based pathology summaries using morphological rules. The

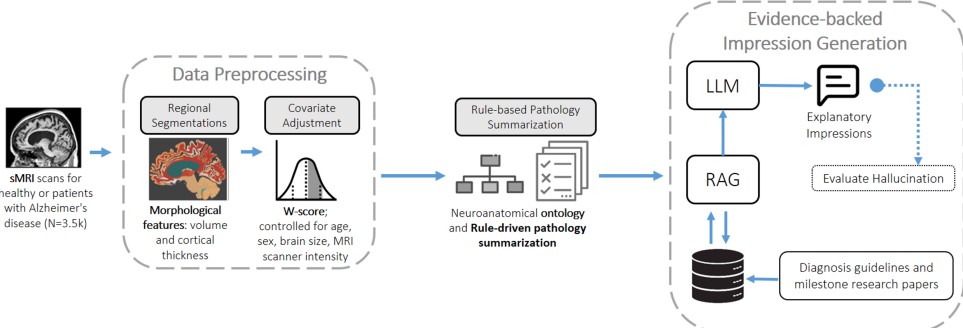

Figure 1: A framework including CNN detection, rule-based templates, and a RAG-grounded LLM for automated guideline-driven impression generation

summary structure was validated by expert radiologists before being used to generate MRI impressions. We assess if the use of a RAG module to ground LLM responses in diagnostic guidelines can improve reliability. Our study evaluates the feasibility of end-to-end MRI-based radiological summary generation and whether a rules-plus-RAG approach produces guideline-consistent diagnostic justifications.

## 2. Methods and Results

We analyzed 3,433 MRIs from multiple cohorts: 47% cognitively normals (CN), 33% with mild cognitive impairment (MCI), 16% with Alzheimer's disease (AD), and 4% with frontotemporal dementia (FTD) (App A). FastSurfer segmented scans into neuroanatomical regions and estimated gray-matter volumes and cortical thickness, which were adjusted for age, sex, and brain size to produce $W$-scores, a covariate-adjusted form of $z$-score reflecting deviation from the expected distribution. A previously developed rule-based system summarized each patient's abnormalities by retaining regions with $W$-scores $> 2\,\mathrm{SD}$ and collapsing subregions into higher-level structures to reduce verbosity, yielding templated text with graded severity (mild/moderate/strong) labels (Singh et al., 2025a). Structured findings were translated into clinically grounded justifications using LLMs via RAG, with prompts engineered to emulate radiologist-style reports. The RAG module was conditioned on four NIA-AA diagnostic recommendation papers (Jack Jr et al., 2024) and the German Neurological Society's (DGN) diagnostic guidelines (DGN e. V. and DGPPN e. V., 2025). The prompts cast the model as a "radiology trainee" to encourage clear, didactic explanations (App. B). We propose a reporting structure where each impression begins by reiterating findings, followed by guideline-based justifications and then lists limitations (App. C). The source code of our framework is available via GitHub[1].

Clinical correctness was evaluated by manually measuring hallucinations in 30 representative patients across disease spectrum. In a micro-averaged analysis, each claim was evaluated for citation presence, information source, and overall claim correctness, yielding seven scenarios: (1) *No hallucination*: cited, RAG-grounded, true claim; (2) *Attribution*

---

1. GitHub repository: https://github.com/DhanushBabu18/MRIs-to-Radiological-Dementia-Reports/

| Hallucination case | AD (N=10) (u=63) | MCI (N=8) (u=43) | CN(N=4) (u=22) | FTD(N=8) (u=37) | Macro Averages |
|---|---|---|---|---|---|
| No hallucinations | 39.7% | 44.2% | 77.3% | 60% | 50.4% |
| Attribution error[2] | 14.3% | 4.7% | 18.2% | 5.4% | 10.3% |
| Contradiction hallucinations | 23.8% | 27.9% | 4.5% | 27.0% | 23.0% |
| Reference hallucinations | 4.8% | 20.9% | 0.0% | 8.0% | 9.1% |
| Extrinsic hallucinations | 0.0% | 2.3% | 0.0% | 0.0% | 0.6% |
| False reference hallucinations | 3.2% | 0.0% | 0.0% | 0.0% | 1.2% |
| Pure fabrication | 14.3% | 0.0% | 0.0% | 0.0% | 5.4% |

*error*: uncited, rag-grounded, true claim; (3) *Contradiction hallucination*: cited, RAG-grounded, false claim (e.g., misinterpretation of the source); (4) *Reference hallucination*: cited, from unknown source, true claim (i.e., fabricated citation); (5) *Extrinsic hallucination*: uncited, from unknown source, true claim; (6) *False reference hallucination*: cited, from unknown source, false claim (i.e., fabricated citation attached to a false claim); (7) *Pure fabrication*: uncited, from unknown source, false claim. Collectively, these categories define our falsifiable hallucination taxonomy for RAG systems (App. D).

We tested several open-source LLMs and performed manual assessments of coherence. Instruction-tuned models best fit our needs, reliably following explicit instructions, whereas medically fine-tuned models under 30B parameters often failed to respond or produced clinically unhelpful outputs (App. E). Among general-purpose models, Mistral-Instruct and Llama-Instruct variants were notably coherent, and Qwen showed strong reasoning but was harder to control. We therefore selected the quantized Llama-3 70B Instruct model for our framework. Using our hallucination taxonomy, we report hallucination rates for representative patients of each disease type and stage, in the table above.

## 3. Discussion

We combine $W$-scores for brain morphology analysis, rule-based neuroanatomical abstraction, and RAG to generate pathological impression reports. Despite grounding the generation in medical guidelines, hallucinations[2] remain common (46% AD, 51% MCI claims). The prevalence of contradictions and reference hallucinations underscores the need to explore more advanced reasoning models and curate more comprehensive RAG sources. Cognitively normal cases showed far fewer hallucinations (4.5%). Existing RAG hallucination mitigation frameworks (Es et al., 2024; Asai et al., 2024), do not systematically characterize how citation behavior, information source, and claim correctness jointly shape hallucination modes (App. F compares it with prior taxonomies). We address this with a RAG-tailored hallucination taxonomy and, to our knowledge, the first end-to-end pipeline for automated radiological impression generation from MRI scans, demonstrating automation potential while underscoring the need for human oversight in using open-scource LLMs.

---

2. Attribution errors were not counted as hallucinations, since the underlying uncited claims still correctly reflected the diagnostic guidelines provided as RAG sources.

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

## Appendix A. Data Cohort Description

In this study, we utilized the normative models developed in our previous work (Singh et al., 2025b), which were trained using T1-weighted volumetric MRI scans from seven data cohorts: the Alzheimer's Disease Neuroimaging Initiative (ADNI; phases ADNI2 and ADNI3), the Australian Imaging, Biomarker and Lifestyle Flagship Study of Ageing (AIBL), the DZNE Longitudinal Study on Cognitive Impairment and Dementia (DELCODE), the European DTI Study on Dementia (EDSD), the DZNE Clinical Registry Study on Frontotemporal Dementia (DESCRIBE-FTD), and the Frontotemporal Lobar Degeneration Neuroimaging Initiative (FTLDNI), also referred to as the Neuroimaging Initiative in Frontotemporal Dementia (NIFD). In total, the study comprised 3,433 MRI scans. Demographic characteristics across diagnostic groups are summarized in Table 1.

Table 1: Demographic and clinical characteristics by diagnostic group. *Values are mean ± SD. CN, cognitively normal; MCI, mild cognitive impairment; AD, Alzheimer's disease dementia; FTD, Frontotemporal dementia; MMSE, Mini-Mental State Examination.*

| Variable | CN (N = 1625) | MCI (N = 1132) | AD (N = 549) | FTD (N = 127) |
|---|---|---|---|---|
| Age (years) | $70.4 \pm 7.6$ | $72.5 \pm 7.4$ | $74.1 \pm 7.7$ | $62.8 \pm 8.2$ |
| MMSE score | $29.1 \pm 1.1$ | $27.6 \pm 2.0$ | $22.2 \pm 4.2$ | $23.7 \pm 5.7$ |
| Education (years) | $15.7 \pm 3.0$ | $15.3 \pm 3.2$ | $13.8 \pm 3.9$ | $14.7 \pm 3.3$ |
| Sex (F/M) | 922 / 703 | 504 / 628 | 280 / 269 | 45 / 82 |

## Appendix B. Prompt Structure

This appendix shows the prompt used (see Fig.2) to generate structured radiology summaries from rule-based template pathology summaries (see Fig.3) as input text. In this setup, the model is constrained to act as a trainee radiology assistant with access only to the diagnostic guideline documents and a clinical significance file (a json file highlighting each regions clinical and functional use). The LLM is instructed to (1) produce a structured FINDINGS section summarizing regional abnormalities and W-scores, (2) generate a guideline-based, multi-paragraph IMPRESSION relating patterns and severity to established biomarker and staging frameworks, and (3) attach explicit filename-based citations to every claim, marking unsupported statements with "Not supported in available sources" and forbidding any use of outside knowledge.

```
prompt=f"""
You are a trainee radiology assistant.

You have access ONLY to the following documents:
1. Alzheimer s Dementia - 2011 - Jack - Introduction to the recommendations from the National Institute on Aging-Alzheimer s.pdf
2. Alzheimer s Dementia - 2011 - McKhann - The diagnosis of dementia due to Alzheimer s disease Recommendations from the.pdf
3. Alzheimer s Dementia - 2011- Albert - The diagnosis of mild cognitive impairment due to Alzheimer s disease.pdf
4. Alzheimer s Dementia - 2018 - Jack - NIA-AA Research Framework Toward a biological definition of Alzheimer s disease.pdf
5. DGN Guidelines Diagnosis.pdf
6. Clinical_Significance.json

If a claim is not explicitly supported in one of these documents, say:
"[Not supported in available sources]".
Do NOT use outside knowledge or make assumptions.

---
PATIENT FINDINGS:
{Patients_findings}
---

TASK:
1. Write **FINDINGS**: summarize abnormalities with W-scores. For each region,specify the pattern, severity, and w-scores, and briefly discuss the potential clinical and pathophysiological relevance, citing one of the six sources by filename.
2. Write **IMPRESSION**:
   - Provide a **multi-paragraph, guideline-based summary**:
   - Describe how the observed atrophy patterns fit into the categories of neuroimaging biomarkers and how the framework classifies such changes .
   - Relate the severity to how the framework discusses early, preclinical, or symptomatic stages.
   - Discuss the pattern and severity of atrophy and its implications for neurodegenerative diseases but do not diagnose any disease.
   - Add a short paragraph explaining the clinical significance about each regions in the findings in detail based on this document Clinical_Significance.json.  If no document supports it, clearly state: "Not supported in available sources."
3. Every claim must be followed by a citation in the format: (Filename, page/section if available).

Do NOT cite anything outside the five listed documents. Do NOT invent facts.

OUTPUT FORMAT:
FINDINGS:
- …

IMPRESSION:
- …
[IMPORTANT]Show step by step before giving the output
[REMEMBER]Only use the documents to claim and do not claim anything outside the documents
"""
```

Figure 2: Prompt template used in the study, constraining the model to predefined diagnostic guidelines while generating structured *findings* and *impression* sections with instructions to cite references.

Moderate pathology in atrophied Left Temporal Lobe (volume w-score: -2.86, CNN relevance w-score: -4.69, cortical_thk w-score: -2.95)
Mild pathology in atrophied Right Inferior Temporal (volume w-score: -2.21, CNN relevance w-score: -2.65 , cortical_thk w-score: -2.20)
Mild pathology in enlarged Left Inf-Lat-Vent (volume w-score: 3.02, CNN relevance w-score: -2.39)

Figure 3: Example of the rule-based template text provided to the LLM. The template comes from a rule-based system that summarizes abnormalities by keeping regions with $W$-scores $> 2\,\mathrm{SD}$ and merging subregions into higher-level structures, producing concise, severity-graded pathology descriptions.

## Appendix C. LLM impression responses for each disease type

Figures 4–7 present LLM-generated impression summaries for representative Alzheimer's Disease (AD), Mild Cognitive Impairment (MCI), Cognitively Normal (CN), and Frontotemporal Dementia (FTD) cases, with claims marked as non-hallucinated (green) or hallucinated (red).

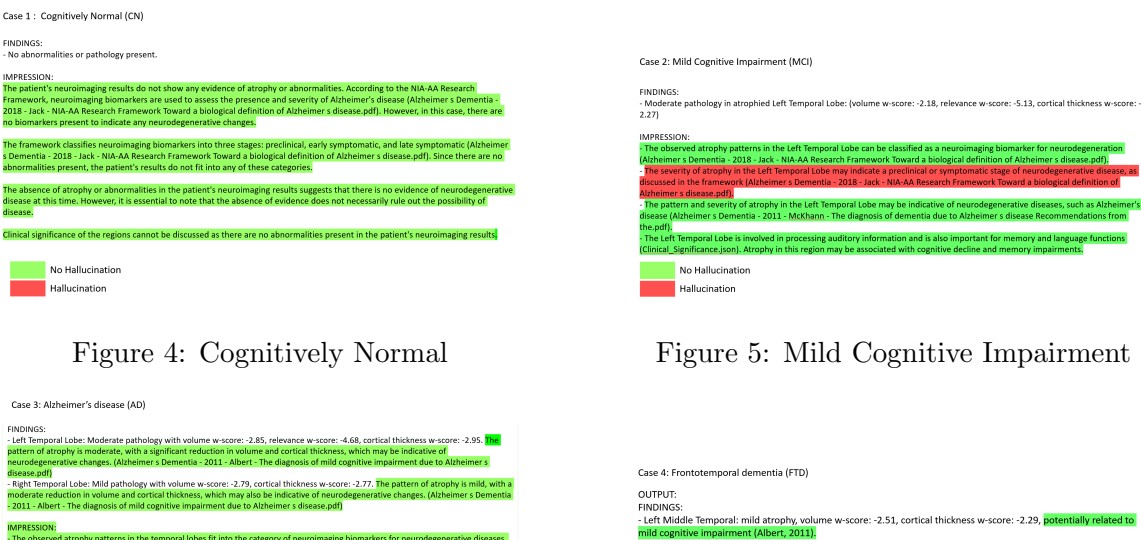

Figure 4: Cognitively Normal

Figure 5: Mild Cognitive Impairment

Figure 6: Alzheimer's Disease

Figure 7: Frontotemporal Dementia

# Appendix D. Hallucination Scenarios

This appendix provides illustrative examples of various hallucination type defined in our falsifiable RAG taxonomy. The following samples demonstrate how citation absence, information source, and claim correctness interact to produce distinct error categories. We highlight non-hallucinated (green) or hallucinated (red) claims. See Figures 8-13.

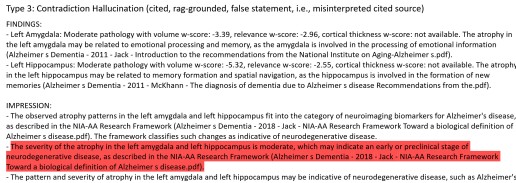

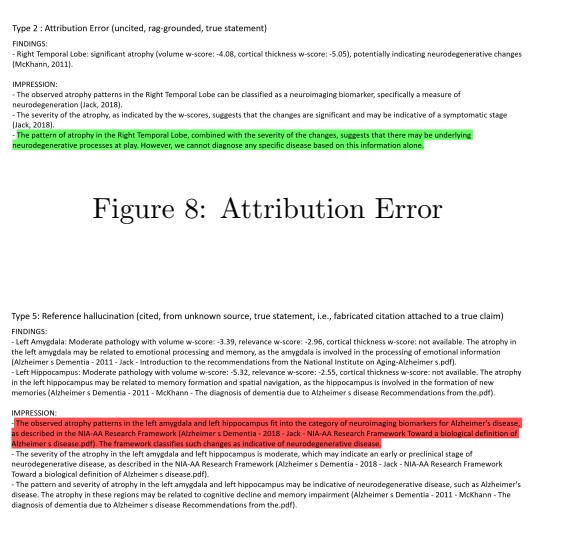

Figure 8: Attribution Error

Figure 9: Contradiction Hallucination

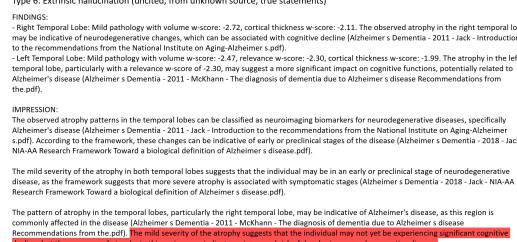

Figure 10: Reference hallucination

Figure 11: Extrinsic hallucination

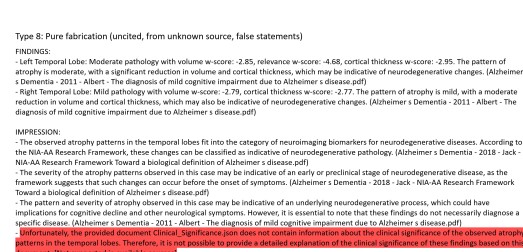

Figure 12: False reference Hallucination

Figure 13: Pure fabrication

## Appendix E. Hallucinations with medically fine tuned LLM

In this section, we present the hallucination rates for Qwen-3 32B Medical Reasoning (4-bit quantized), a medically fine-tuned large language model (See Table 2). These results allow us to evaluate how domain-specific tuning affects reliability across multiple failure modes.

For context, we compare these outcomes to those of the Llama-3 70B Instruct model, which we selected as our default baseline. Relative to this baseline, Qwen-3 32B Medical Reasoning shows a substantial increase in hallucination rates. For the same set of representative patients, hallucinations[3] rose to 52.4% (a 6% increase) for AD patients, 68.9% (a 17.8% increase) for MCI patients, and 72.3%, representing the largest increase of 37.7%, for FTD patients. These findings indicate that, contrary to our expectations, the medical fine-tuning did not yield performance improvements for our use case. Instead, the results suggest that model size remains a critical determinant of reliability, and in our experience, larger-parameter models consistently perform better.

Table 2: Measured hallucination rates for Qwen-3 32B Medical Reasoning LLM. *N denotes sample size; u denotes evaluated claims.*

| Hallucination case | AD (N=10) (u=63) | MCI (N=8) (u=43) | CN(N=4) (u=22) | FTD(N=8) (u=37) | Macro Averages |
|---|---|---|---|---|---|
| No hallucination | 46.0% | 29.5% | 70.0% | 27.7% | 40.8% |
| Attribution error[3] | 1.6% | 1.6% | 30.0% | 0.0% | 5.0% |
| Contradiction hallucination | 42.9% | 24.6% | 0.0% | 30.8% | 29.7% |
| Reference hallucination | 1.6% | 36.1% | 0.0% | 32.3% | 17.3% |
| Extrinsic hallucination | 4.8% | 3.3% | 0.0% | 1.5% | 3.0% |
| False reference hallucination | 0.0% | 0.0% | 0.0% | 1.5% | 0.3% |
| Pure fabrication | 3.2% | 4.9% | 0.0% | 6.2% | 3.9% |

---

3. Attribution errors were not counted as hallucinations, since the underlying uncited claims still correctly reflected the diagnostic guidelines provided as RAG sources

## Appendix F. Validating Hallucination Taxonomy

To contextualize and validate our hallucination taxonomy, we compare it with the framework proposed by Huang et al (Huang et al., 2025). Their survey offers a broad categorization of hallucinations in LLMs, largely assuming a stand-alone generation setting. In contrast, our taxonomy defines hallucinations as falsifiable scenarios arising under RAG, yielding a more operational framework for high-risk, evidence-grounded applications.

Several of Huang et al.'s categories map directly onto our taxonomy. Their *contradiction errors* correspond to our *contradiction* and *contextual* hallucinations, while *factual fabrication* decomposes into *false reference hallucinations* and *pure fabrication*, depending on citation behavior. This correspondence indicates conceptual convergence, with our framework resolving these errors into RAG-specific, operationally distinct cases.

A key conceptual difference lies in the treatment of extrinsic hallucinations. While Huang et al. acknowledge that LLMs may generate true statements sourced outside the provided context, this case is not included in their final taxonomy. Our framework explicitly models uncited, externally sourced true claims as grounding violations and further distinguishes reference hallucinations, where correct statements are paired with fabricated citations—an error type not addressed by Huang et al. These distinctions show that factual correctness alone is insufficient for evaluating corpus-grounded generation.

Our taxonomy also has some limitations. Our taxonomy focuses exclusively on factual grounding under RAG and does not explicitly measure faithfulness hallucinations, such as instruction inconsistency or logical incoherence as response relevance metrics, which are addressed in prior work (Huang et al., 2025; Es et al., 2024). Instead, these aspects were handled indirectly through prompting strategies and rule-based constraints in our pipeline. Additional limitations include reliance on time-consuming, manual claim-level evaluation and residual ambiguity in uncited statements, where distinguishing RAG-grounded content from pretrained knowledge remains challenging.

In summary, our taxonomy complements prior hallucination frameworks by offering a RAG-specific, falsifiable view of grounded generation errors, while highlighting open challenges in modeling instruction-level and reasoning-based failures.

