# OpenReview forum: "Evidence-Supported Automated Impressions for Alzheimer’s Disease Detection from Brain MRI: A Feasibility Study"
_MIDL.io/2026/Short_Papers — MIDL 2026 - Short Papers Poster_

### Official Review · Reviewer_6wVN · 2026-04-23
**analysis of hallucinations in MRI LLM**

**Rating:** 5
**Confidence:** 3

**Review:**

The paper’s main strengths are its clinical relevance, its sensible hybrid design combining quantitative MRI analysis with rule-based abstraction and guideline-grounded LLM generation, and its careful claim-level evaluation of hallucinations rather than relying on overall impressions. It is also valuable that the authors report a negative but important result: even with RAG and rule-based guardrails, hallucinations remain substantial, which makes the study useful for understanding the current limits of automated reporting. The main weakness is that the detailed evaluation is based on a small manually reviewed sample of 30 cases but this is acceptable for a short paper.

**Summary:**

The paper presents a pipeline that converts quantitative brain MRI abnormalities into radiology-style impressions for dementia assessment using rule-based summaries plus a guideline-grounded LLM. The main contribution is in experimenting with a constrained, auditable setup and showing how it still fails. The authors also propose a claim-level hallucination taxonomy tailored to RAG, separating errors by truthfulness, source grounding, and citation behavior.

**Strengths:**

- automated, evidence-backed MRI impression generation for dementia is an interesting use case or LLMs in the medical domain.
- hybrid design combining quantitative morphology, rule-based abstraction, and RAG-grounded LLM generation.
- claim-level hallucination evaluation is more informative than generic output-level judgments.
- even with guardrails, hallucinations remain high, which is valuable observation for the field.
- the taxonomy is practically useful for auditing grounded generation

**Weaknesses:**

- Hallucination analysis is somewhat limited (30 cases) which limits the generalizability of the findings from this study to some extent. On the other hand, I find that this is acceptable for a short paper.

**Justification Of Rating:**

The empirical result that hallucinations remain high is important for the clinical translation of LLM medical report generation.

---

### Decision · Program_Chairs · 2026-05-08

Accept (Poster)